# Image Compression and Classification Using Qubits and Quantum Deep Learning

## Abstract

Recent work suggests that quantum machine learning techniques can be used for classical image classification by encoding the images in quantum states and using a quantum neural network for inference. However, such work has been restricted to very small input images, at most $4 \times 4$, that are unrealistic and cannot even be accurately labeled by humans. The primary difficulties in using larger input images is that hitherto-proposed encoding schemes necessitate more qubits than are physically realizable. We propose a framework to classify larger, realistic images using quantum systems. Our approach relies on a novel encoding mechanism that embeds images in quantum states while necessitating fewer qubits than prior work. Our framework is able to classify images that are larger than previously possible, up to $16 \times 16$ for the MNIST dataset on a personal laptop, and obtains accuracy comparable to classical neural networks with the same number of learnable parameters. We also propose a technique for further reducing the number of qubits needed to represent images that may result in an easier physical implementation at the expense of final performance. Our work enables quantum machine learning and classification on classical datasets of dimensions that were previously intractable by physically realizable quantum computers or classical simulation.

## 1 Introduction

In the past decade, deep learning has been remarkably successful on a wide variety of classical learning tasks (Gi (2016); LeCun et al. (2015)). In parallel, quantum computing (QC) has long promised dramatic increases in computational power over classical computers, culminating in a recent demonstration of quantum supremacy in a machine with 53 programmable qubits in Arute et al. (2019). However, even these quantum systems are already approaching the limits of classical simulability by the world's largest traditional supercomputers (Boixo et al. (2018); Preskill (2012)). The power of quantum computation suggests that quantum analogues of deep learning models like feedforward neural network may outperform their classical counterparts, especially when the data is inherently quantum (Wan et al. (2017); Beer et al. (2020); E. Farhi (2018)).

In this paper, we use a quantum neural network (QNN) to classify the MNIST dataset of handwritten digits (LeCun & Cortes (2010)). Prior work has been restricted to only highly-compressed, rather unrealistic input images due to their inefficient encoding schemes that are injective maps from classical images to their corresponding pure quantum states. These frameworks have used input images of a maximum resolution of $4 \times 4$, which is too coarse even for humans to provide accurate labels (see Figure 1). For larger images, injectivity would necessitate the same number of qubits as bits that are present in the original image. However, recent advances in quantum machine learning (QML) on classical data, such as the Flexible Representation of Quantum Images (FRQI) in P.Q. Le & Hirota (2011), have demonstrated that quantum wavefunctions can utilize quantum entanglement to encode *classical* data using exponentially less qubits than their corresponding classical representation in bits. In this paper, we use the FRQI method to embed the input images in fewer-qubit systems. This approach necessitates a novel QNN architecture for classification, which we describe in Section 5. The input to our model are images of resolution up to $16 \times 16$ whose quantum encoding only requires 8 qubits (6 for pixel locations, 1 for color, and 1 for readout). To the best of our knowledge, our work is the first to propose a data encoding scheme and QNN that can be used to classify *realistic* images.

**Our main contributions are as follows:**

1. We provide a novel construction to compress images and encode them in their FRQI states. Our construction uses only 2-qubit gates, which permits its use in common quantum machine learning packages such as Cirq and Tensorflow Quantum (TFQ) and may be of independent interest (Cirq (2021); GoogleAI (2020)).

2. We propose a new QNN layers, CRADL and CRAML, which we use in a model trained with the images' FRQI states as input.

3. We show that our trained QNN achieves accuracy comparable to classical models with the same number of parameters.

4. We propose a novel technique to further compress black and white images, and study the scaling behavior of our model with the extent of image compression.

**Organization:** In Section 2, we provide a brief review of the formalism of quantum computation. In Section 3, we provide an overview of related work and motivate our study. In Section 4, we describe our dataset and how each image is encoded in a quantum state. In Section 5, we describe the prescription for using a quantum neural network to obtain a classification prediction for a given image and describe the model we use for classification. In Section 6, we present our results. We conclude with Section 7 and a description of future work in Appendix 8.

## 2 PRELIMINARIES

### 2.1 QUANTUM COMPUTING

Here we provide a brief review of quantum computation. For a more detailed reference and an interactive coding tutorial, we refer the reader to Nielsen & Chuang (2011) and Qiskit (2017) respectively.

In quantum computation, the basic unit of information is a two-state quantum mechanical (QM) system called a qubit; the two states are traditionally written $|0\rangle$ and $|1\rangle$. A qubit can be in either of these two states, as well as a quantum superposition of these states, formally written as a wavefunction $|\psi\rangle = a_0 |0\rangle + a_1 |1\rangle = \sum_{i \in \{0,1\}} a_i |i\rangle$, where each $a_i \in \mathbb{C}$. When a qubit is measured, the wavefunction collapses and the result of the measurement is state $|0\rangle$ with probability $|a_0|^2$ and state $|1\rangle$ with probability $|a_1|^2$ [1]. The space of all possible states of the qubit is called the Hilbert space $\mathcal{H}_1$; the states $|0\rangle$ and $|1\rangle$ provide a basis for this Hilbert space. [2]

Multi-qubit systems are represented mathematically by the tensor product of multiple single-qubit systems. Notationally, we write

$$|\psi_N\rangle = \sum_{\{i_1, i_2, \dots i_N\} \in \{0,1\}^N} a_{i_1, i_2, \dots i_N} |i_1, i_2, \dots i_N\rangle$$

where the states $|i_1, i_2, \dots i_N\rangle$ provide a basis for the multi-qubit Hilbert space $\mathcal{H}_N$ and $|\psi_N\rangle$ is generally a superposition of these basis states. A two-qubit state $|\psi\rangle \in \mathcal{H}_2$, cannot necessarily be factorized into two single-qubit states $|\psi_1\rangle, |\psi_2\rangle \in \mathcal{H}_1$:

$$|\psi\rangle = \sum_{\{i_1, i_2\} \in \{0,1\}^2} a_{i_1, i_2} |i_1, i_2\rangle \neq |\psi_1\rangle \otimes |\psi_2\rangle = (a_0 |0\rangle + a_1 |1\rangle) \otimes (a'_0 |0\rangle + a'_1 |1\rangle)$$

we call a state which cannot be so factored a mixed state. In particular, we notice that $\mathcal{H}_2 > \mathcal{H}_1 \otimes \mathcal{H}_1$, and a similar result holds for multi-qubit systems with $N > 2$.

Under the laws of quantum mechanics, these wavefunctions – or states – evolve in time as determined by linear unitary transformations. Furthermore, any operation that is physically possible to perform

---

[1] We require $|a_0|^2 + |a_1|^2 = 1$

[2] Formally, a Hilbert space is an inner product vector space that is also a complete metric space with respect to the distance function induced by that inner product.

on a set of qubits can be represented as a unitary operator. We refer to such unitary operators as quantum gates and note that unitary operators can be viewed as rotations in Hilbert space.

A remarkable property of quantum states is their ability to be entangled. Informally, entanglement refers to the property of quantum mechanical systems whereby the state of one qubit cannot be described independently of the other qubits' states. For example, the state $|00\rangle$ is maximally entangled as knowledge of one qubit's state complete specifies the other's.

In general, quantum algorithms are procedures whereby an initial wavefunction is transformed under a sequence of unitary operations, or quantum gates, and a measurement is made of the transformed state; this measurement is often performed on a readout qubit and is the output of the algorithm. Many quantum computation algorithms are designed to exploit properties of entangled systems (Bernstein & Vazirani (1997); David & Richard (1992); Simon (1997); Shor (1997); Grover (1996)).

## 2.2 COMMON QUANTUM GATES

Several common quantum gates are defined below. These gates are defined by their action on the basis states of the Hilbert space, since they extend linearly to superpositions of the basis states. The definitions for the single-qubit Hadamard gate $H$ and the Pauli-$X$ gate $X$ are:

$$H|0\rangle = |+\rangle := \tfrac{1}{\sqrt{2}}(|0\rangle + |1\rangle) \qquad H|1\rangle = |-\rangle := \tfrac{1}{\sqrt{2}}(|0\rangle - |1\rangle) \tag{1}$$

$$X|0\rangle = |1\rangle \qquad\qquad\qquad X|1\rangle = |0\rangle \tag{2}$$

A common 2-qubit gate is $CNOT$, which flips the second qubit if the first qubit is in state $|1\rangle$:

$$CNOT|00\rangle = |00\rangle \qquad CNOT|01\rangle = |01\rangle \tag{3}$$
$$CNOT|10\rangle = |11\rangle \qquad CNOT|11\rangle = |10\rangle$$

## 3 RELATED WORK

Many studies use QNNs to model either inherently quantum or quantum-encoded classical data but are generally restricted to very small images (Li et al. (2020); Henderson et al. (2020); Oh et al. (2021)). One line of work encodes classical data in quantum systems and focuses on learning the classifier's circuit architecture. These approaches require an injective map from the input image to a corresponding pure quantum state, which forgoes the exponential compression advantages afforded by methods such as FRQI[3] (E. Farhi (2018); Aïmeur et al. (2013); Paparo et al. (2014); Schuld et al. (2014); Kapoor et al. (2016)). Amongst this line of work, E. Farhi (2018) propose the general setup that we follow in this paper. In contrast with their work, however, we use the FRQI technique to exploit the dimensionality of the multi-qubit Hilbert space and need much fewer qubits.

Other studies take the quantum wavefunction as given, either by assuming the classical data is already provided in its quantum-encoded form Schuld et al. (2020) or because they use inherently quantum data (Sasaki & Carlini (2002); Gambs (2008); Sentís et al. (2012); Dunjko et al. (2016); Monràs et al. (2017); Alvarez-Rodriguez et al. (2017); Du et al. (2020); Sentís et al. (2019); Beer et al. (2020); Caro & Datta (2020)). Amongst these papers, Schuld et al. (2020) is perhaps closest to this work. The authors, however, take the mixed-state encodings of images as given for input to a QNN and do not describe how to construct the quantum states. Other work, such as Beer et al. (2020), assumes the wavefunctions as given proposes a generalization of the perceptron to the quantum setting, which provides a more generalized framework than in E. Farhi (2018) These authors use inherently quantum data, in contrast with our work. We use classical data and explicitly construct quantum circuits to encode *classical* images into their wavefunctions. Our approach lends itself to direct experimentation and is usable with modern quantum machine learning packages.

Finally, a third line of work uses quantum convolutional neural networks via semi-classical simulations meant to model the noise introduced by quantum effects, as in Kerenidis et al. (2019). These approaches do not provide a fully quantum simulation to evolve the quantum states, which would require construction of the actual data's wavefunctions as in our work.

---

[3]For a demo of a standard approach following E. Farhi (2018) using TFQ, which could also serve as a preliminary for this work, see https://www.tensorflow.org/quantum/tutorials/mnist

Throughout prior work, encoding classical data in quantum states *efficiently* appears to be a common open problem.

## 4 FORMAL SETTING

### 4.1 PROBLEM STATEMENT

In classical image classification, the input to our model is an $n \times n$-pixel image $\in \{0, 1\}^{2n}$ and our goal is to learn a classification function with binary outputs $f_{\text{classical}}$ parametrized by weights $w$:

$$f_{\text{classical}}(w) : \{0, 1\}^{2n} \to \{0, 1\} \tag{4}$$

In the quantum setting, the input to our classification function is still an $n \times n$-pixel image but must be encoded in a $\lceil \log 2n \rceil + 1$ dimensional Hilbert space $\mathcal{H}$ by an encoding function $\mathcal{F}$, where the $+1$ is for the readout qubit. The quantum neural network is a sequence of unitary operations $\mathcal{U} = U_1 \circ U_2 \circ \ldots U_N$ parametrized by angles $\theta = \theta_1, \theta_2, \ldots, \theta_N$. To obtain a classification prediction, a measurement is performed on the readout qubit:

$$f_{\text{quantum}}(\theta) : \{0, 1\}^{2n} \to \{0, 1\} \tag{5}$$

$$: \{0, 1\}^{2n} \xrightarrow{\mathcal{F}} \mathcal{H} \xrightarrow{\mathcal{U}(\theta)} \mathcal{H} \xrightarrow{\text{measure}} \{0, 1\}$$

That is $f_{\text{quantum}}(\theta) = \text{measure} \circ \mathcal{U}(\theta) \circ \mathcal{F}$.

In Sections 5 and 5 we propose an implementation of the FRQI algorithm P.Q. Le & Hirota (2011) to construct $\mathcal{F}$, propose a construction of $U(\theta)$, describe how to learn the parameters $\theta$ via standard backpropagation, and describe the final measurement step.

### 4.2 DATASET AND QUANTUM ENCODING

Crucial to our approach is the encoding of a classical datapoint (e.g. an image) in a quantum state. In our experiments, we use the MNIST dataset of handwritten digits LeCun & Cortes (2010). Following E. Farhi (2018), we restrict our dataset to those of only two ground truth labels: 3 and 6. We downsample image resolutions to either $8 \times 8$ or $16 \times 16$ using bilinear interpolation. The remaining dataset is approximately $12,000$ training images and $1,100$ validation images for each resolution. Finally, we transform the images to black and white by thresholding the pixel color.

In Figure 1, we present an MNIST image downsampled to different resolutions. Prior work uses resolutions of only $4 \times 4$, but loses many important features of the original data E. Farhi (2018). With the FRQI encoding and the further compression we are able to encode higher-resolution images on current quantum hardware; this insight motivates our study of different downsampled resolutions.

After preprocessing, each image is a black and white $2^n \times 2^n = 2^{2n}$ dimensional binary vector. Our objective is to encode the image as a wavefunction $|\psi_{\text{data}}\rangle$:

$$|\psi_{\text{data}}\rangle = \sum_{q := \{q_0, q_1, \ldots, q_{2n-1}\} \in \{0,1\}^{2n}} |q_0, q_1, \ldots, q_{2n-1}\rangle \otimes (\cos \theta_q |0\rangle + \sin \theta_q |1\rangle) \tag{6}$$

In Equation 6, each basis state $|q_0, q_1, \ldots, q_{2n-1}\rangle$ of the "pixel qubits" represents a possible bitstring of length $2^{2n}$ with the strength of the superposition component and color determined by $\theta_q$ in the "color qubit". In our dataset, each $\theta_q$ is either $0$ or $\frac{\pi}{2}$.

In some experiments in Section 5, we also consider allocating more qubits to encode the color angle $\theta_q$ instead of the pixel locations:

$$|\psi_{\text{data}}\rangle = \sum_{q := \{q_0, q_1, \ldots, q_{2n-1}\} \in \{0,1\}^{2n}} |q_0, q_1, \ldots, q_{2n-3}\rangle \otimes \left( \cos \tilde{\theta}_q |0\rangle + \sin \tilde{\theta}_q |1\rangle \right)$$

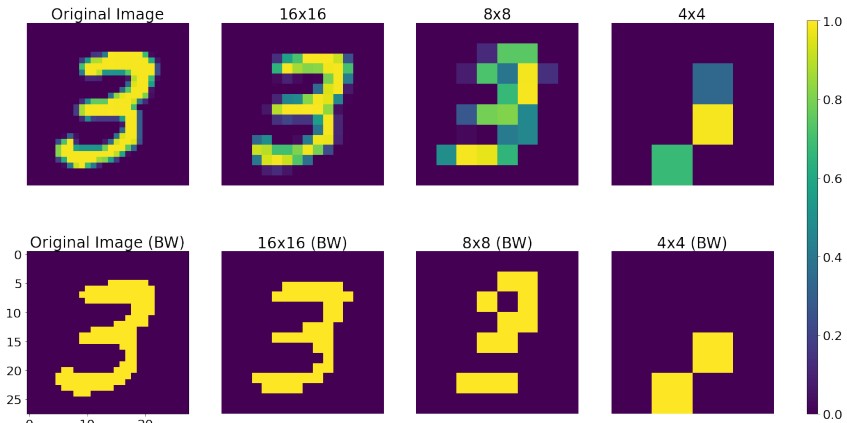

Figure 1: A digit from MNIST presented at different downsampled resolutions (downscaled resolutions indicated on top of each image. The top row consists of grayscale images with $0 \leq$ color $\leq 1$ as in the original dataset. The bottom row are black and white images obtained by thresholding the pixel color from the respective images above.

To do this, we exploit the observation that the color qubit is always either $|0\rangle$ or $|1\rangle$ and map into it the last two pixel qubits according to the transformation:

$$|q_{2n-2}, q_{2n-1}\rangle \otimes |q_c\rangle \rightarrow |\tilde{q}_c\rangle = \cos\tilde{\theta}_q |0\rangle + \sin\tilde{\theta}_q |1\rangle \tag{7}$$

where each $q_i \in \{0, 1\}$ and

$$\tilde{\theta}_q = \frac{\pi}{2}\left(q_c + \frac{q_{2n-2}}{2} + \frac{q_{2n-1}}{4}\right) = \theta_q + \frac{\pi}{4}\left(q_{2n-2} + \frac{q_{2n-1}}{2}\right) \tag{8}$$

# 5 METHODS

## 5.1 ENCODING THE IMAGES IN WAVEFUNCTIONS

In our approach, we must first encode the image in a quantum wavefunction. We pass an initial state of $|0\ldots0\rangle$ through a quantum circuit with a given structure, demonstrated by Figure 2 for a 4-qubit state. Initially, a Hadamard operation $H^{\otimes 2n}$ is performed on the $2n$ pixel qubits. This is followed by a series of $n$-controlled $X$-gates (also known as generalized TOFFOLI gates Toffoli (1980); Rasmussen et al. (2020); Shende & Markov (2008)) with alternating $X$ gates that determine the color qubits which will be transformed. The $n$-qubit circuit is constructed recursively from smaller-qubit circuits by observing the symmetries in the construction.

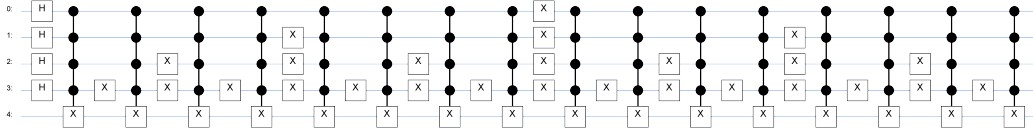

Figure 2: An example circuit to construct the superposition state for 4 qubits, using 4-qubit generalized TOFFOLI gate. We follow standard quantum circuit diagram conventions in which the dots represent the control of a gate by the given qubits.

To construct this circuit we need to define the generalized TOFFOLI gate for $n$ qubits from basic two-qubit gates; this also enables our implementation in standard packages such as Cirq (Cirq (2021)) that only support backpropagation through two-qubit gates. We use the following lemma, first shown in Barenco et al. (1995), to recursively decompose the $n$-qubit generalized TOFFOLI gate as a sequence of $(n-1)$-qubit generalized TOFFOLI gates and CNOT gates:

**Lemma 1.** *(Barenco et al. (1995), Lemma 7.5): For a rotation matrix $R(t)$, an $n$-controlled rotation gate can be decomposed into a circuit of the form shown in Figure 3.*

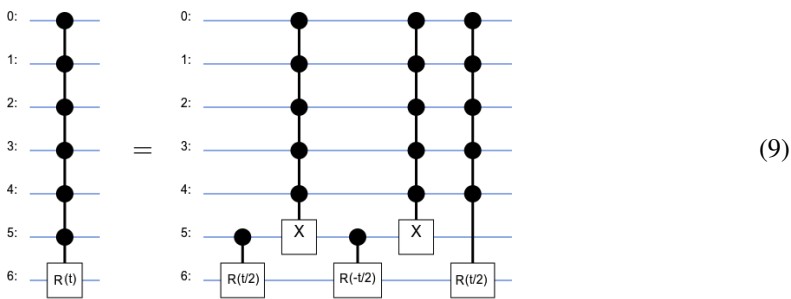

$$(9)$$

Figure 3: $n-$qubit controlled gate in terms of $(n-1)-$qubit controlled gates

From the recursive properties of the diagram above we see that an $n$-qubit controlled rotation decomposes into $(2 \cdot 3^n - 1)$ one-qubit controlled rotations. Since we flip at most $2^n$ pixels for a given image, we see that for each image

$$\text{\# one-qubit controlled gates } \leq 2^n \left(2 \cdot 3^n - 1\right)$$

in addition to the standalone $X-$gates to encode the $2^n \times 2^n$ image in the amplitudes of the input wavefunction. Though this bound is exponential in $n$, we find this acceptable as it is still classically simulable for larger images that are primarily constrained by the size of their qubit representations.

## 5.2 READOUT QUBIT AND PREDICTED LABELS

Our wavefunction must also contain a readout qubit on which we perform measurements that will be the model's predicted labels. As such, we prepare the wavefunction $|\psi_{\text{in}}\rangle = |\psi_{\text{data}}\rangle \otimes |\text{readout}\rangle$. We choose the $Z$-gate for measurement and thus initialize the readout qubit in the $|+\rangle$ state, which is common practice to produce an initially unbiased output:

$$|\psi_{\text{in}}\rangle = |\psi_{\text{data}}\rangle \otimes H\,|0\rangle = |\psi_{\text{data}}\rangle \otimes |+\rangle \tag{10}$$

where $|\psi_{\text{data}}\rangle$ is prepared as in Section 5.1.

The model which is used to transform $|\psi_{\text{in}}\rangle$ is a QNN with $L$ layers. Following E. Farhi (2018), each layer is represented by a parametrized unitary matrix. The model's output state is:

$$|\psi_{\text{out}}(\theta)\rangle = U(\theta_L)\ldots U(\theta_1)\left(|\psi_{\text{data}}\rangle \otimes |+\rangle\right) \tag{11}$$

where $\theta \coloneqq (\theta_1, \ldots, \theta_L)$. The final measurement is performed with $Z-$gate on the readout qubit; the predicted label is $\langle\psi_{\text{out}}(\theta)|\,\mathbb{I}^{2n+1} \otimes Z\,|\psi_{\text{out}}(\theta)\rangle$. We train the model's parameters, $\theta_1 \ldots \theta_L$, via stochastic gradient descent (SGD) using these predictions and the hinge loss:

$$\text{loss}^{(i)}(\theta) = 1 - y^{(i)}\,\langle\psi_{\text{out}}^{(i)}(\theta)|\,\mathbb{I}^{2n+1} \otimes Z\,|\psi_{\text{out}}^{(i)}(\theta)\rangle \tag{12}$$

where the superscript $(i)$ is used to refer to the $i^{th}$ training example.

## 5.3 IMPLEMENTATION DETAILS

We use Cirq (Cirq (2021)) to encode the images into their respective wavefunctions and TensorFlow Quantum (TFQ) (GoogleAI (2020)) to train the model via the paradigm described in Section 5.2. TFQ permits the use of Parametrized Quantum Circuits (PQCs), which describe the unitary operations of the QNN, as a single Keras layer Chollet et al. (2015) within the standard TensorFlow framework.

Backpropagation through quantum layers is nontrivial. We recall that, for any layer, the $2^n \times 2^n$ unitary operator can be expressed in terms of the exponential of a $2^n \times 2^n$ Hermitian operator $H$ called the Hamiltonian, which in turn can be decomposed into its Pauli decomposition (a tensor product of $n$ Pauli matrices, which form an orthonormal basis over the Hilbert space of Hermitian matrices over $\mathbb{R}$):

$$U(\theta) = \exp \left\{ i \sum_{\sigma_i \in \{\mathbb{I}, \sigma_1, \sigma_2, \sigma_3\}^n} \theta^{(\sigma_i)} \bigotimes_i \sigma_i \right\} \tag{13}$$

When the layer is restricted to unitary operations whose Hamiltonian has a single term in the Pauli decomposition, gradients with respect to the layer's parameters can be computed analytically using the parameter shift techniques first introduced in Mitarai et al. (2018) and Schuld et al. (2019) (Harrow & Napp (2021)). These techniques provide a way to calculate partial derivatives of parameterized quantum circuits in terms of other functions that use the same circuit architecture with shifted parameters. For this reason, we restrict the gates in our quantum layers to be multi-qubit exponential Pauli gates. For these gates, analytic gradients can also be computed because they are rotations of operations whose Hamiltonians contain a single type of term. For example, the $XX$-gate can be written:

$$(X \otimes X)^\theta = \exp \left\{ \theta \left( -i \tfrac{\pi}{2} (X - \mathbb{I}) \otimes -i \tfrac{\pi}{2} (X - \mathbb{I}) \right) \right\} = e^{-i \frac{\pi}{2} \theta (X - \mathbb{I})} \otimes e^{-i \frac{\pi}{2} \theta (X - \mathbb{I})}$$

### 5.4 NETWORK ARCHITECTURE

A general $2^n \times 2^n$ learnable unitary operation would consist of $2^{2n}$ trainable real parameters and the standard representation of this parameters follows equation equation 13. Note that, as described in Section 5.3, this does not necessarily permit analytic gradient computation in the backpropagation step.

To permit analytical gradient computations, we construct layers having a specific structure; an example such layer is demonstrated in Figure 4. Each layer consists of either $XX$ or $ZZ$ operators applied in succession to each pixel qubit and the readout qubit, followed by the same operation on the pixel qubit and the color qubit. Empirically, the Color-Readout-Alternating-Double-Layer architecture (CRADL) presented in Figure 4 resulted in the best performance.

Each consecutive pixel-readout and pixel-color pair of gates share the same learning parameter, a rotation angle. One double layer of the type shown in Figure 4 will have $2n$ learnable parameters, where $n$ is the number of pixel qubits. A circuit with $L$ layers will therefore have $nL$ parameters. In the experiments in Section 6, we choose the number of layers such that the number of parameters $nL$ is comparable to those of the classical benchmark, given a fixed number of qubits.

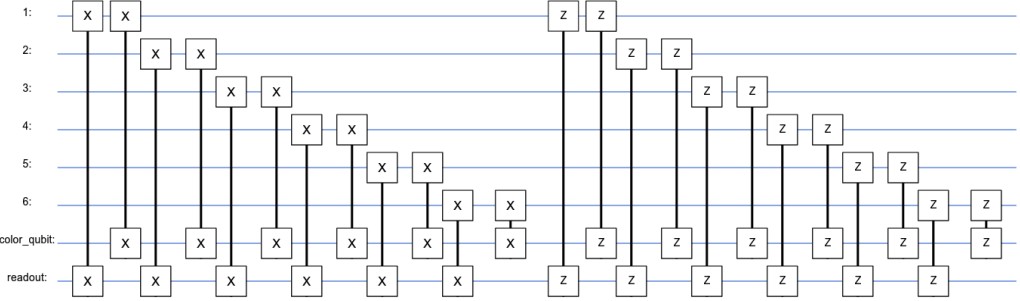

Figure 4: A "CRADL" network double layer with 6 pixel qubits, consisting of consecutive pixel-readout pixel-color $XX$ gates, followed by analogous $ZZ$ gates

We note that there are equivalent network architectures that lead to comparable results, such as the Color-Readout-Alternating-Mixed-Layer architecture shown in Figure 5.

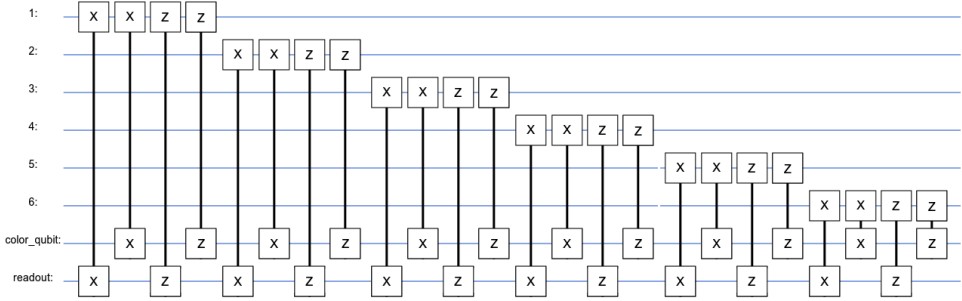

Figure 5: A "CRAML" network layer with 6 pixel qubits, consisting of consecutive pixel-readout pixel-color $XX$ and $ZZ$ gates

## 6 RESULTS

We benchmark our quantum learning framework against a classical neural network with two hidden layers with ReLU activations and a single-neuron output layer; in this setting, the quantum and classical models have a comparable number of parameters. We trained the quantum neural network for 10 epochs, which is the same number of epochs after which the classical neural network began to overfit (as determined by cross-validation). All experiments were conducted on a personal laptop with no GPU (Macbook Pro, 2.4 GHz 8-Core Intel Core i9 CPU, 64 GB 2667 MHz DDR4 RAM).

| Network | $8 \times 8$ **Image** | $16 \times 16$ **Image** |
|---|---|---|
| Classical CNN | $94 \pm 1\%$ | $98.9 \pm 0.3\%$ |
| Quantum CRADL | $92 \pm 1\%$ | $-$ |
| Quantum CRADL $-2Q$ | $88 \pm 1\%$ | $90 \pm 1\%$ |

Table 1: Test accuracies after the 10th training epoch for classical and quantum networks.

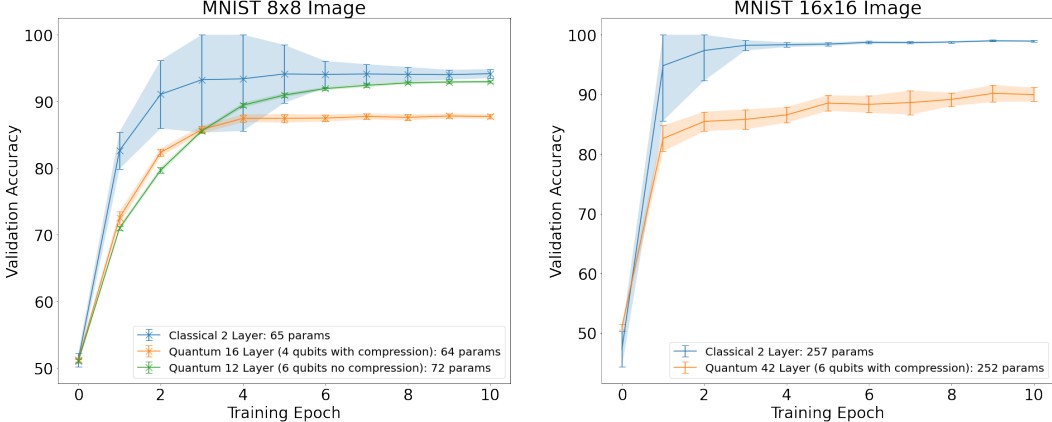

Figure 6: Test accuracy versus training epoch for classical and quantum models, with and without the extra 2 qubit compression described in Section 4.2. When images are embedded on 6 qubits (green curve, left), we achieve performance comparable to classical networks with the same number of parameters. When images are further compressed (orange curves), performance degrades.

Table 1 demonstrates our final results. On $8 \times 8$ images, the QNN without the extra two-qubit compression achieves performance comparable to the classical network, whereas the network with the extra two-qubit compression (denoted $-2Q$) performs worse. For $16 \times 16$ images, we add more layers to the QNN so that the the number of parameters remains comparable to the classical dense network and must use the extra two-qubit compression due to the computational cost of the experiment. We

observe that the QNN is unable to achieve the same performance as the classical network, likely due to the extensive compression of the images in the quantum states.

Figure 6 shows the validation accuracy of our quantum and baseline classical models versus training epoch. We observe that the classical neural network and both quantum neural networks demonstrate similar validation performance curves.

## 7    DISCUSSION AND CONCLUSION

We note that the method we describe in Equation 8 to reduce the number of required qubits results in worse performance. This degradation in test accuracy may be palatable in applications that attempt to minimize their qubit usage. When we attempted to lower the number of necessary qubits further, we observed unstable learning behavior. In such settings with reduced feature dimensionality, it may be necessary to redesign the network architecture.

In this paper, we developed a proof of concept for recently proposed QNN models. In the process, we proposed a methodology to map classical images to quantum states that may be of independent interest to the community. We also propose a new form of quantum neural network layers motivated by the highly entangled input states, the CRADL and CRAML layers in Section 5.3, and demonstrate that a model consisting of these layers achieves performance competitive with classical neural networks with a comparable number of parameters. Furthermore, our work is evidence that quantum machine learning algorithms can scale to data of dimensions larger than those previously tractable by classical simulation or available quantum hardware and classify MNIST images of size $16 \times 16$ on a personal laptop.

## 8    FUTURE WORK

We did not do a comprehensive survey of the space of all possibly unitary operations that could be used for each hidden layer, but we could imagine that invoking 3 or more qubit gates to the layer circuits would improve the learning outcomes, as that would give more direct access to the correlation structure. The circuit above Figure 4 composed entirely of two-qubit gates is trying to work around this limitation.

We would like in the future to conduct a more systematic survey of network architectures to enhance the learning outcomes with possibly more involved quantum gates, either within TensorFlow Quantum or implemented separately, and study the cost benefit of forgoing analytic gradients for some of these gates for more flexibility in construction. Another lower hanging fruit is optimizing the encoded input circuits in terms of memory. As long as we write and simulate quantum algorithms on digital computers, representing the wavefunction in this manner seems inevitable and we should find better optimizations to describe the encoding procedure through circuits with much less gates than the nice recursive algorithms discussed in this paper.

We observed in Section 6 that compressing the images into larger quantum systems resulted in better performance at the expensive of greater complexity in physical realization. We leave a more thorough analysis of this tradeoff, including the understanding of the interplay between the data qubits and the color qubit, to future work.

We also note that we may view the encoding with limited circuit gates or on limited qubits as a form of implicit regularization, which has been observed to improve generalization performance as in Smith et al. (2016). We leave investigation of these connections to future work.

## ETHICS STATEMENT

We do not foresee any ethical concerns with our work.

## REPRODUCIBILITY STATEMENT

The authors have submitted all the code necessary to reproduce their results in the Supplementary Materials as a `.zip`. The code contains a `README.md` file with instructions on how to reproduce the experimental results. The only dataset required is the MNIST dataset, which is easily obtained from `http://yann.lecun.com/exdb/mnist/`. The preprocessing applied to the MNIST dataset is described in detail in Section 5.

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

## SUPPLEMENTARY MATERIAL AND APPENDIX

Supplementary code submitted as a `.zip`.

