# OpenReview forum: "Image Compression and Classification Using Qubits and Quantum Deep Learning"
_ICLR.cc/2022/Conference — ICLR 2022 Submitted_

### Official Review · Reviewer_PiSA · 2021-10-30

**Correctness:** 4
**Technical Novelty And Significance:** 2
**Empirical Novelty And Significance:** 2
**Recommendation:** 3
**Confidence:** 5

**Main Review:**

From my perspective, the topic of image classification with practical implementation studied by this paper is interesting and can be of general impact. However, after taking a closer look at this paper, I feel that the contribution in results as well as the novelty in algorithm design are not significant for this paper to be accepted by ICLR 2022. Please find my detailed comments below:

- To me, the technical contribution of this paper is minor. In particular, although the encoding from Eq. (6)-(8) was not claimed in previous papers on image classification, the math is standard with only one extra qubit added. For the circuit synthesis in Section 5, it is mostly the standard trick in Lemma 1 proposed by Barenco et al. more than 20 years ago, and for the circuit implementation the authors simply call the standard package TensorFlow Quantum from Google AI. The Flexible Representation of Quantum Images (FRQI) was also originally proposed by Le and Hirota back in 2011. In general, compared to the previous work of Farhi and Neven which tests 4*4 input images on quantum neural networks, this work is more like running experiments with moderately larger encodings and scales, but the methods are routine.

- I feel that the abstract has significant overclaims compared to what the paper achieves. In particular, the abstract claimed that their result “obtains accuracy comparable to classical neural networks with the same number of learnable parameters”. However, in Section 6 of the paper, the authors admit that “we observe that the QNN is unable to achieve the same performance as the classical network”. Looking at Table 1, the classical success probability is 98.9%+-0.3% while quantum is 90+-1%, both in 10 epochs; the quantum result is actually much worse than the classical counterpart, not even comparable to me. In addition, it is also claimed that “our work enables quantum machine learning and classification on classical datasets of dimensions that were previously intractable by physically realizable quantum computers or classical simulation”, which is not wrong but does not make much sense to me because classical computers can run images with resolution much much better than 16*16. In addition, the authors’ experiments are restricted to only numbers 3 and 6. In all, also with my first point above, this work is not only still a toy model compared to the current scale of classical machine learning methods, but also does not offer much value to the design of quantum machine learning algorithms.

- Although the paper is not long, it has various typos:

- - At the end of Section 1: “a description of future work in Appendix 8” -> “a description of future work in Section 8”

- - Section 3, related work: The reference to E. Farhi (2018) should be changed to Farhi and Neven (2018) as in https://arxiv.org/pdf/1802.06002.pdf

- - Eq. (4) and the text beforehand: I believe the definition domain should change from {0,1}^{2n} to {0,1}^{n^2}

- - End of Section 4.1, Sections 5 and 5 should be fixed; for the reading of the same sentence, P.Q. Le & Hirota (2011) -> (Le & Hirota (2011)). In many places of the paper there should be brackets for references.

**Summary Of The Paper:**

This paper studies the image classification problem on quantum computers. The prior of image classification by Farhi et al. can only work on 4-by-4 input images, while this paper conducts experiments on 16-by-16 images for the MNIST dataset.

**Summary Of The Review:**

Based on the concerns I mentioned above, I recommend rejection for this paper at ICLR 2022.

---

### Official Review · Reviewer_snts · 2021-11-02

**Correctness:** 2
**Technical Novelty And Significance:** 2
**Empirical Novelty And Significance:** 2
**Recommendation:** 5
**Confidence:** 2

**Main Review:**


#### Positive
- The related work and formal problem description sections are well written and organized.
- The experimental validation of the proposed approach goes beyond single runs.

#### Negative
- The introduction is extremely vague. Why would quantum computers outperform their classic counterparts? What does it mean for data to be "inherently quantum"?

- Authors claim to have the ability to 8x8 or 16x16 encode images on current quantum hardware. A least one major company now offers free access to their machines.
    - Are any experiments on actual quantum hardware available?

- If "these quantum systems are already approaching the limits of classical
simulability by the world’s largest traditional supercomputer" (page 1), why would simulation experiments on a consumer laptop suffice in the paper's case?

- Page 5 observes that the color qbit is either $\braket{0}$ or $\braket{1}$.
  MNIST digits are black and white. Would this observation hold for color images?

#### Minor remarks
page 3, top: .. completely specifies the other's.

**Summary Of The Paper:**

The paper presents an image coding approach based on 2qubit gates.
Downsampled MNIST digits are classified using standard components.
Additionally, the experimental section studies the use of reduced codes.

**Summary Of The Review:**

The paper presents an interesting idea, however, since the new contributions appear to be limited to equations seven and eight, I am not convinced the contributions are sufficient for inclusion in the proceedings.

---

### Official Review · Reviewer_N7sU · 2021-11-02

**Correctness:** 3
**Technical Novelty And Significance:** 1
**Empirical Novelty And Significance:** 1
**Recommendation:** 1
**Confidence:** 5

**Main Review:**

Strength:
1. Use FRQI to embed images with higher resolution.
2. Propose CRADL and CRAML as parameterized circuits to perform the classification tasks.

Weakness:
1. Major: The novelty seems limited, the embedding method which encodes the images to the state vector of the qubits, is a well-established one and has already been used in previous QNN work such as [1]. The proposed parameterized circuit part is also similar to that in [2].
2. Major: Although the image resolution can be large, from 4x4 to 8x8 or 16x16, they are binarized, which will obviously lose a considerable amount of information. Why binarized 8x8 is guaranteed to be better than int 4x4 such as that used in [3]?
3. Major: since the paper claims to have a good embedding method, comparisons with other embedding methods are necessary, such as phase-based embedding [3][4]; state-vector embedding[2], etc.
4. Major: The proposed embedding circuit seems to be very deep (figure 2), which will be significantly impacted by noise in NISQ device. Some experiments on real quantum devices, such as on publicly available IBMQ machines can help justify the results.
5. Major: The experiment is only performed on a single MNIST dataset and there are no ablation studies on the hyperparameters it used. Comparisons with previous QNN work are completely missing.
6. Minor: the claim of "our work is the first to propose a data encoding scheme and QNN that can be used to classify realistic
images." is overclaimed, [2][3][5] all perform classification on the realistic MNIST dataset. [3] even perform 10-class and 4-class classifications on real quantum machines.
7. Minor: As in figure 6, the empirical results seem not good enough as the classical NN model can consistently outperform the quantum models.
8. Minor: More justifications on why using the CRAML and CRADL should be provided. The design space is very large, why choose those kinds of ansatz?




[1] Jiang, Weiwen, Jinjun Xiong, and Yiyu Shi. "A co-design framework of neural networks and quantum circuits towards quantum advantage." Nature communications 12.1 (2021): 1-13.

[2] Farhi, Edward, and Hartmut Neven. "Classification with quantum neural networks on near term processors." arXiv preprint arXiv:1802.06002 (2018).

[3] Wang, Hanrui, et al. "Quantumnas: Noise-adaptive search for robust quantum circuits." arXiv preprint arXiv:2107.10845 (2021).

[4] Lloyd, Seth, et al. "Quantum embeddings for machine learning." arXiv preprint arXiv:2001.03622 (2020).

[5] Henderson, Maxwell, et al. "Quanvolutional neural networks: powering image recognition with quantum circuits." Quantum Machine Intelligence 2.1 (2020): 1-9.


**Summary Of The Paper:**

The paper study the problem of quantum neural networks for classical image classification. It leverages the FRQI framework to enable larger size of the input image instead of using very small 4x4 input image. They construct parameterized quantum circuits using XX and ZZ gates (called "CRADL" and "CRAML") to perform the transformations of quantum state and perform classification.

**Summary Of The Review:**

The novelty is insufficient as the embedding part and parameterized circuit part is not new. the evaluation is insufficient: no comparisons with state-of-the-art; no evaluation on real quantum devices; no ablation studies; no justifications of why the proposed embedding can outperform others.

---

### Official Review · Reviewer_MiCP · 2021-11-05

**Correctness:** 4
**Technical Novelty And Significance:** 2
**Empirical Novelty And Significance:** 2
**Recommendation:** 3
**Confidence:** 4

**Main Review:**

The paper provides a novel quantum algorithm for loading and classifying images inspired from classical machine
learning models. The authors try to incorporate several properties of quantum computing into their algorithm. Given an
image, the authors start by downsampling it to a lower resolution and build a quantum circuit that loads this image into a
quantum state. This data-loading circuit is made with elementary gates that build a superposition of pixel colors. An
additional readout register is added to the circuit where trainable gates by gradient descent are added to the overall circuit
in order to map the input to its label. Moreover, they also run small experiments on two different resolutions (8x8,16x16)
and show how their model compares to a classical CNN.
However, the paper has several drawbacks for acceptance at ICLR. The contributions are very marginal compared to
state-of-art results in quantum machine learning. First, the data-loading procedure assumes that a binary representation
of the image is enough for classification which may not be true for datasets with low-contrast images. The number of
qubits is logarithmic with respect to the image dimensions but the circuit seems to have a large depth. The authors do not
compare their procedure to other data-loaders used in quantum machine learning. Second, the quantum circuit used for
classification is neither novel nor close to state-of-art quantum neural networks. It is not clear what this circuit is able to
learn and what are its strengths and limits. Last, the paper lacks a theoretical analysis of the proposed quantum algorithm
except some arguments provided in section 5.1. It is not clear what are the gate and time complexities and how does it
compare to existing approaches

**Summary Of The Paper:**

In this paper, the authors propose a certain type of trainable quantum procedures used as machine learning models for
the classification of images. They explicitly show how to implement the circuits for loading compressed images into
quantum states that are fed to a trainable unitary that writes the predicted label into a readout register. They conclude on
numerical simulation performed on the MNIST dataset.

**Summary Of The Review:**

The authors proposed a new algorithm for classification problems combining a new quantum data loader for images and
an existing trainable quantum circuit. However, the paper lacks arguments that motivate using their new approach
because the authors did not provide neither empirical evidence that their quantum algorithm outperforms existing
approaches nor improvement of the running time to train their circuit

---

### Author Response · Authors · 2021-11-20
**Acknowledging the work needed to qualify our paper for publication**

We agree that our paper requires substantial improvements to meet the standards for publication at ICLR, and appreciate the time the reviewers have spent to provide thoughtful feedback. We will use this to improve our submission in the future

---

### Decision · Program_Chairs · 2022-01-20

**Decision:**

Reject

**Comment:**

This submission proposes a new encoding mechanism, i.e. a new quantum data loader for images with a reduced number of qubits, which is then used for image classification with off-the-shelf quantum neural networks (from TensorFlow Quantum). There are two major concerns raised by most reviewers. The first concern regards the novelty in the design of the quantum data loader and the use of off-the-shelf quantum neural networks (QNNs), the latter of which is neither novel nor close to state-of-art QNNs for the same purpose.  The quantum data loading procedure also assumes a binary representation of images that might not be enough for low-contrast images. Although the number of qubits is reduced, the circuit tends to have a large depth which makes it hard for practical implementations.  The second concern regards the overall performance of the proposed solution for image classification, where a clear quantum benefit is missing, or in some cases, a quantum disadvantage shows up.  Based on these discussions, we believe that the submission requires substantial improvements before its publication.